# VARIATIONAL RECURRENT MODELS FOR REPRESENTATION LEARNING

## ABSTRACT

We study the problem of learning representations of sequence data. Recent work has built on variational autoencoders to develop variational recurrent models for generation. Our main goal is not generation but rather representation learning for downstream prediction tasks. Existing variational recurrent models typically use stochastic recurrent connections to model the dependence among neighboring latent variables, while generation assumes independence of generated data per time step given the latent sequence. In contrast, our models assume independence among all latent variables given non-stochastic hidden states, which speeds up inference, while assuming dependence of observations at each time step on all latent variables, which improves representation quality. In addition, we propose and study extensions for improving downstream performance, including hierarchical auxiliary latent variables and prior updating during training. Experiments show improved performance on several speech and language tasks with different levels of supervision, as well as in a multi-view learning setting.

## 1 INTRODUCTION

Modeling sequence data is a central problem in domains such as speech and natural language processing. In this work, we study the problem of learning representations (features) of sequence data, with varying degrees of supervision, that can be helpful for downstream sequence labeling tasks. We take a generative approach, inspired by variational autoencoders (VAEs) Kingma & Welling (2014); Rezende et al. (2014); Doersch (2016) for non-sequence data. We assume that the sequence data is generated by a series of latent variables, parameterize the observation likelihood with recurrent neural networks, and maximize (a lower bound on) the observation likelihood. If labels are available during training, a discriminative loss can also be imposed on the latent variables. The learned posterior distribution of the latent variables then provides features for the downstream tasks. This intuitively simple approach, however, requires non-trivial extensions of the deep generative models that have been successful for non-sequence data.

For a (non-sequence) observation $x$, a VAE assumes a prior distribution, $p(z)$, on the latent variable $z$ which is often simple (e.g., $\mathcal{N}(0, I)$), and consists of a generation network parameterizing $p(x|z)$ with weights $\theta$ and an inference network parameterizing the approximate posterior $q(z|x)$ with weights $\phi$. Learning the VAE is done by maximizing an evidence lower bound (ELBO) on the observation log-likelihood $\log p_\theta(x) = \log \int p(z) p_\theta(x|z) dz$, defined as

$$ELBO := \mathbb{E}_{q_\phi(z|x)} \left[ \log p_\theta(x|z) \right] - D_{KL} \left( q_\phi(z|x) || p(z) \right) \leq \log p_\theta(x). \tag{1}$$

VAEs can be viewed as autoencoders: The encoder $q(z|x)$ maps the input to a latent variable $z$ according to a (also typically Gaussian) posterior distribution, and the decoder $p(x|z)$ reconstructs the input from samples of $z$ drawn from this posterior. The output of the encoder (most commonly, the mean of $q(z|x)$) is often used as a learned representation (features) for downstream tasks. Maximizing the ELBO is equivalent to minimizing the reconstruction loss of the output $x$ with respect to the input, plus a regularizer based on the prior distribution of the latent variable. In some variants the importance of the KL term is tuned by weighting the regularizer by a hyperparameter $\beta$ Higgins et al. (2016); Alemi et al. (2016).

VAEs have been extended to model sequence data in a number of ways. Fabius & van Amersfoort (2014) learn a single representation for the entire sequence, while Hsu et al. (2017) learn both a

whole-sequence representation and a set of representations for pre-defined segments within the sequence. For many tasks, such as the ones we consider here, it is desirable to represent a length-$T$ input sequence $x_{1:T}$ with a corresponding length-$T$ latent sequence $z_{1:T}$ so as to fit directly into typical recurrent network-based prediction models.

Several recent approaches fit this criterion Krishnan et al. (2015); Archer et al. (2015); Chung et al. (2015b); Fraccaro et al. (2016); Goyal et al. (2017a); Chen et al. (2018). Learning is again done by maximizing an ELBO, with the main differences among approaches being the specific forms of the prior $p(z_{1:T})$ (typically parameterized so as to capture dynamics in the latent space), the generation distribution $p_\theta(x_{1:T}|z_{1:T})$, and the approximate posterior $q_\phi(z_{1:T}|x_{1:T})$. For example, direct recurrent connections between stochastic variables Fraccaro et al. (2016), as shown in Figure 1(a), or indirect recurrent connections, e.g. $h_{t-1} \to z_{t-1} \to h_t \to z_t$ Chung et al. (2015b); Goyal et al. (2017a), are often introduced in $p(z_{1:T})$ to model the dependence between neighboring latent variables. While this is more powerful than a simpler prior for the purpose of generation, it poses challenges for designing the approximate posteriors due to the dependencies among $z_t$'s. On the other hand, given $z_{1:T}$, the generation model is often fully factorized into $T$ independent terms: $p_\theta(x_{1:T}|z_{1:T}) = \prod_{t=1}^{T} p_\theta(x_t|z_t)$.

Most prior work on variational recurrent models has focused on generation quality and likelihood evaluation. It is not clear, however, that the learned representations are useful for downstream tasks. The goal of our work is to fill this gap by developing variational recurrent models that produce high-quality representations for prediction tasks, and to investigate several modeling choices.

We take an approach, similarly to Chen et al. (2018), where the latent distribution $p(z_{1:T})$ factors over time conditioned on a *deterministic* hidden state sequence, while the generation model $p_\theta(x_{1:T}|z_{1:T})$ is factored as $\prod_{t=1}^{T} p_\theta(x_t|z_{1:T})$, as shown in Figures 1(b,d). Also similarly to Chen et al. (2018), we use a simple Gaussian prior which is updated during training, and find that this can be very helpful. Unlike Chen et al. (2018), however, we consider a wide variety of tasks and levels of supervision. Compared to variational recurrent models with recurrent connections among the stochastic latent variables, this class of models is more efficient and easier to implement, since there are no dependencies between stochastic latent variables and therefore no nested sampling procedure.

One key novelty in this work is that, while we have no recurrent connections among the $z_{1:T}$, we require *each $z_j$ to generate each $x_k$* with some probability, for all $1 \le j, k \le T$, so that the features are predictive of a large context of observations. We propose a "stochastic generation" model that approximates this idea, and we find that this extension is important for improving downstream performance.

In addition, we study the proposed models on a range of speech and NLP prediction tasks, in a variety of supervision settings, finding improved performance in all cases. We consider fully supervised, semi-supervised, and weakly supervised (using multi-view training data) settings. We believe this is the first attempt at a broad study of recurrent variational representation learning that is applicable to a variety of downstream tasks and levels of supervision.

## 2 A VARIATIONAL RECURRENT REPRESENTATION LEARNING MODEL

Our variational recurrent representation learning approach, shown in Figure 1(b-d), aims to learn informative representations while keeping inference simple. As in other recurrent variational models, we optimize an ELBO of the same form as in Equation 1 but where the variables are all sequences of length $T$; that is, the prior is $p(z_{1:T})$, the posterior is $q_\phi(z_{1:T}|x_{1:T})$, and the generation model is $p_\theta(x_{1:T}|z_{1:T})$.

### 2.1 INFERENCE

Given a length $T$ input $x_{1:T}$, the model first maps from the input sequence to a sequence of deterministic recurrent hidden state vectors $h_{1:T}$, produced by one or more stacked recurrent neural network layers. Conditioned on $h_t$ ($1 \le t \le T$), the posterior distribution of the latent random variable $z_t$ is a spherical Gaussian

$$z_t|h_t \sim q_\phi(z_t|h_t) \equiv \mathcal{N}(\mu_{\phi_t}, \mathrm{diag}\,(\sigma_{\phi_t})), \quad [\mu_{\phi_t}, \log \sigma_{\phi_t}] = F_\phi(h_t) \qquad (2)$$

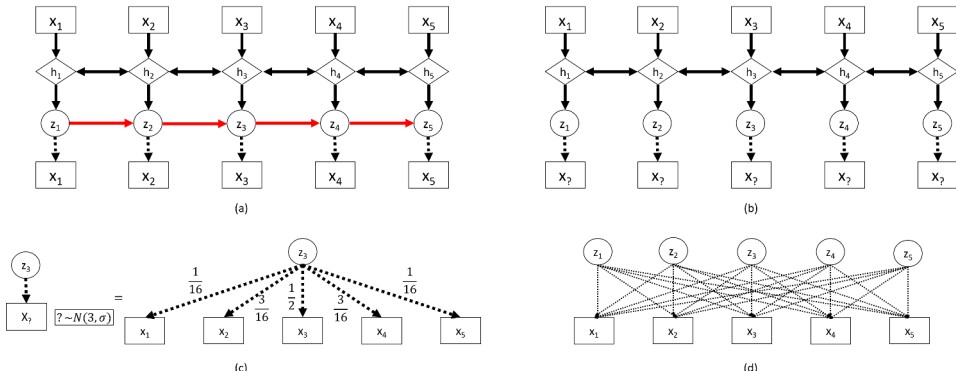

Figure 1: a) A variational recurrent model with stochastic recurrent connections (red arrows); (b) Proposed model with no latent stochastic variables dependencies and with stochastic generation; (c) An example of stochastic generation, with a particular choice of Gaussian stochastic generation distribution; (d) The complete generation process of our model, where each $x_k$ is generated with some probability by each $z_t$. Throughout, dashed lines indicate generation given samples of latent variables, and double-headed arrows indicate bidirectional recurrent connections. In (a,b), only one layer of deterministic hidden states $h_t$ is shown, but any number can be used.

Here, $q_\phi$ is the inference network with parameters $\phi$ and $F_\phi$ is either a linear transformation or a feedforward neural network with a linear final layer. The full posterior and prior are

$$q_\phi(z_{1:T}|h_{1:T}) := \prod_{t=1}^{T} q_\phi(z_t|h_t), \quad p(z_{1:T}) := \prod_{t=1}^{T} p(z_t) \tag{3}$$

Finally, a reconstruction of the input sequence is generated from the latent stochastic sequence $z_{1:T}$. The learned representation of $x_t$, which can be used for downstream tasks, is taken to be the mean $\mu_{\phi_t}$ of $q_\phi(z_t|h_t)$.

Note that no direct dependence is assumed among different time steps of $z_{1:T}$ given $h_{1:T}$. This greatly simplifies inference over prior work that directly models dependence between the $z_t$ Goyal et al. (2017a); Fraccaro et al. (2016); Chung et al. (2015b); Krishnan et al. (2015), since the distribution of each $z_t$ (given $h_t$) is a single spherical Gaussian rather than a more complex multimodal distribution that would arise with dependence modeling (e.g., using recurrent connections among stochastic nodes). Consider a model with stochastic recurrent connections as in Figure 1(a), which models the conditional probability of $z_t$ given the past $z_{1:t-1}$. The marginal distribution of each $z_t$ is multimodal, and requires nested sampling. In order to obtain good estimates, a large number of samples of $z_{1:t-1}$ may need to be used. In addition, in a model with multimodal marginal distributions of $z_t$, it is not clear what to consider to be the learned representation for use in a downstream tasks; in particular the mean of $z_t$ may have very small probability and thus may not be a good choice.

## 2.2 STOCHASTIC GENERATION

Whereas our approach simplifies inference, the generation of each $x_k$ for $1 \leq k \leq T$ in our model involves samples of all $z_{1:T}$, with each $z_t$ for $1 \leq t \leq T$ having a different effect on the generation of $x_k$, as shown in Figures 1(c,d). Intuitively, this generation approach reintroduces some of the relationships between the $z_t$ that may have been lost by having no direct dependence modeling. Given a particular choice of $k, t$, the generative model for $x_k$ given $z_t$ is also a spherical Gaussian:

$$x_k|z_t \sim \mathcal{N}(\mu_{\theta_t}, diag(\sigma_{\theta_t})) = p_\theta(x_k|z_t), \quad [\mu_{\theta_t}, \log \sigma_{\theta_t}] = F_\theta(z_t) \tag{4}$$

where again $F_\theta$, with parameters $\theta$, can be linear or a feedforward neural network with a linear final layer. Note that the full generative distribution (Figure 1(d)) is multimodal, although we never explicitly evaluate it, as described below.

Figure 1(d) shows our full generation process. However, this generation process is a bit expensive; instead we employ *stochastic generation* (Figure 1(c)), a technique that aims to approximate the generation process. In stochastic generation, we first randomly select a frame $x_k$ from $x_{1:T}$ for time step $t$. During training we evaluate $\mathbb{E}_{q_\phi(z_t|h_t)}\big[\log p_\theta(x_k|z_t)\big]$ for time step $t$. The way we sample the target to be reconstructed for time step $t$ affects the way $z_t$ and $h_t$ encode $x_{1:T}$. The higher the probability that a distant time frame is selected, the more temporal information should be encoded in the latent variables. Next we describe stochastic generation more formally.

**Definition 2.1.** $\alpha_{\delta,T}^{(t,k)}$ Given $x_{1:T}$ and Gaussian distribution $p^{(t)}(s) \equiv \mathcal{N}(t,\delta)$, where $1 \le t \le T$. Let $c_{0:T} = \{-\infty, 1.5, 2.5, ..., T-0.5, +\infty\}$. We define $\alpha_{\delta,T}^{(t,k)}$ for $1 \le t, k \le T$ as

$$\alpha_{\delta,T}^{(t,k)} := \int_{c_{k-1}}^{c_k} p^{(t)}(s)ds \tag{5}$$

**Definition 2.2. Stochastic Generation** Given $\delta$ and sequence $x_{1:T}$, stochastic generation reconstructs $x_k$ using samples of $z_t$ with probability $\alpha_{\delta,T}^{(t,k)}$. That is, the conditional log-likelihood for time step $t$ is $\mathbb{E}_{q_\phi(z_t|h_t)}\big[\log p_\theta(x_k|z_t)\big]$ with probability $\alpha_{\delta,T}^{(t,k)}$.

In stochastic generation, each $z_t$ contributes to $x_k$ differently based on the delay between time steps $t$ and $k$. If $\delta$ is chosen such that only nearby neighbor $x_k$s are produced, it becomes a weighted window generation. Remark 1 shows how a particular instance of stochastic generation relates to a full generation model, with graphical model as described in Figure 1(d).

*Remark* 1. Given $z_{1:T}$, $\phi$ and $\theta$, stochastic generation in expectation is evaluating

$$\sum_{t=1}^{T} \left\{ \mathbb{E}_{q_\phi(z_t|h_t)} \left[ \sum_{k=1}^{T} \alpha_{\delta,T}^{(t,k)} \log p_\theta(x_k|z_t) \right] \right\} \tag{6}$$

Equation 6 is equivalent to computing the expectation of $\log p_\theta(x_{1:T}|z_{1:T})$, where

$$p_\theta(x_{1:T}|z_{1:T}) = \prod_{t=1}^{T} p_\theta(x_t|z_{1:T}) = \prod_{t=1}^{T}\prod_{k=1}^{T} p_\theta(x_k|z_t)^{\alpha_{\delta,T}^{(t,k)}} \tag{7}$$

It is straightforward to show that the ELBO from the two perspectives is identical (see the supplementary material). Through stochastic generation, although we are always using simple Gaussian distributions, we are implicitly creating a more complex generative model. Note that, for purposes of generation and likelihood evaluation, we directly use the generative model described in Equation 7. More detailed derivations can be found in the supplementary material.

## 3 EXTENSIONS

Thus far we have described a basic recurrent variational representation learning model that uses stochastic generation to account for context instead of recurrent stochastic connections. We next describe a few extensions aimed at improving the learned representations for use in downstream tasks. First, we embed our model in a multitask training approach, in order to utilize different levels and types of supervision (full supervision, semi-supervision, and weak supervision in a multi-view learning setting). Next, similarly to Chen et al. (2018), we consider models with a hierarchy of latent variables per time step and specify a different functionality for each one, which allows some of the latent variables to focus on encoding task-specific/view-invariant information; and we study the use of prior updating in order to improve the quality of inference with our factored posterior.

### 3.1 LEARNING WITH SUPERVISION

If we have a task-specific prediction model with loss $\mathcal{F}$ and labels $l$, then we can maximize the following objective

$$(1-\alpha)\big\{\mathcal{L}_{\delta,\beta}(x_{1:T}, z_{1:T}, \theta, \phi)\big\} - \alpha\mathcal{F}(z_{1:T}, l) \tag{8}$$

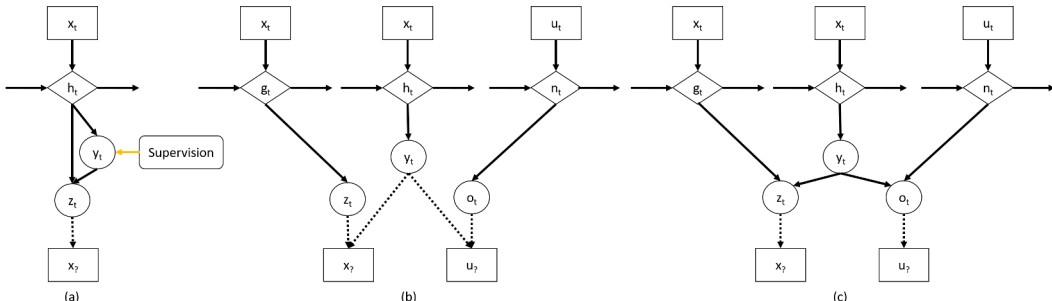

Figure 2: a) A model with hierarchical latent variables. The additional latent variable $y_t$ is task-specific. b) A multi-view recurrent variational model, RecVCCAP. This is a recurrent extension of variational canonical correlation analysis with private variables (VCCAP, Wang et al., 2016). c) The hierarchical version of b, RecVCCAP+H.

where $\mathcal{L}_{\delta,\beta}$ is our representation learning loss (ELBO), with a given stochastic generation parameter $\delta$ and KL term weight $\beta$, and $\alpha > 0$ is a trade-off parameter. This is a multitask loss, where the latent variable $z_t$ needs to perform well with respect to the unsupervised loss (reconstruction of the input and similarity to the prior) as well as the supervised task-related loss.

In some settings labels are not available, but a second view of the data—that is, a second input sequence $u_{1:T}$ paired with each primary input sequence $x_{1:T}$—is available at training time but not test time. For example the second view may be from another modality. In such a setting it is often possible to learn a better representation of the primary view by using the second view as a form of weak supervision Ngiam et al. (2011); Srivastava & Salakhutdinov (2012); Sohn et al. (2014); Wang et al. (2015). Here, we extend the variational canonical correlation analysis with private variables (VC-CAP) model of Wang et al. (2016) to a variational recurrent multi-view model with stochastic generation, named RecVCCAP for short.

In RecVCCAP, a latent representation $y_{1:T}$ is inferred from $x_{1:T}$ only, but used to reconstruct both modalities. In addition, $z_{1:T}$ conditioned on deterministic hidden state $g_{1:T}$, and also $o_{1:T}$ conditioned on deterministic $n_{1:T}$, are also introduced to model information specific ("private") to $x$ and $u$ respectively, as shown in Figure 2(b). The ELBO of this model is a lower bound on the joint distribution of data in the two views:

$$
\begin{aligned}
&\log p_\theta(x_{1:T}, u_{1:T}) \\
\geq\ & E_{q_\phi(z_{1:T}, y_{1:T}|g_{1:T}, h_{1:T})}\big[\log p_\theta(x_{1:T}|z_{1:T}, y_{1:T})\big] + E_{q_\phi(o_{1:T}, y_{1:T}|n_{1:T}, h_{1:T})}\big[\log p_\theta(u_{1:T}|o_{1:T}, y_{1:T})\big] \\
-\ & D_{KL}\big(q_\phi(o_{1:T}|n_{1:T})||p(o_{1:T})\big) - D_{KL}\big(q_\phi(z_{1:T}|g_{1:T})||p(z_{1:T})\big) - D_{KL}\big(q_\phi(y_{1:T}|h_{1:T})||p(y_{1:T})\big) \\
=\ & \Sigma_{t=1}^T \bigg\{ \mathbb{E}_{q_\phi(y_t|h_t)q_\phi(z_t|g_t)}\Big[\Sigma_{k=1}^T\big\{\alpha_{\delta,T}^{t,k}\log p_\theta(x_k|z_t)\big\} + \mathbb{E}_{q_\phi(y_t|h_t)q_\phi(o_t|n_t)}\Sigma_{k=1}^T\big\{\alpha_{\delta,T}^{t,k}\log p_\theta(u_k|o_t)\big\}\Big]\bigg\} \\
-\ & D_{KL}\big(q_\phi(o_{1:T}|n_{1:T})||p(o_{1:T})\big) - D_{KL}\big(q_\phi(z_{1:T}|g_{1:T})||p(z_{1:T})\big) - D_{KL}\big(q_\phi(y_{1:T}|h_{1:T})||p(y_{1:T})\big) \quad (9)
\end{aligned}
$$

Here the generation model includes $p_\theta(x_{1:T}|z_{1:T}, y_{1:T})p(z_{1:T})p(y_{1:T})$ and $p_\theta(u_{1:T}|o_{1:T}, y_{1:T})p(o_{1:T})p(y_{1:T})$ for the two modalities respectively. Detailed derivation of the ELBOs can be found in supplementary material.

## 3.2 Hierarchical latent variables

In supervised training, it may be beneficial to separate the latent variables into task-specific and task-independent components, and to apply the supervised loss only to the task-specific latent variables. The multitask model of Equation 8 can be easily extended with an auxiliary task-specific latent variable $y_t$, as shown in Figure 2(a). The supervised loss is applied only to $y_t$ while the unsupervised loss (reconstruction + prior) is applied to $z_t$. The multitask loss (negative ELBO+supervised loss)

of this hierarchical model is

$$-(1-\alpha)\big\{\mathcal{L}_{\delta,\beta}(x_{1:T}, z_{1:T}, y_{1:T}, \theta, \phi)\big\} + \alpha\mathcal{F}(y_{1:T}, l)$$

$$\equiv -(1-\alpha)\Sigma_{t=1}^{T}\bigg\{\mathbb{E}_{q_{\phi}(y_t|h_t)q_{\phi}(z_t|y_t,h_t)}\bigg[\Sigma_{k=1}^{T}\big\{\alpha_{\delta,T}^{(t,k)}\log p_{\theta}(x_k|z_t)\big\}\bigg]$$

$$- \beta\big\{D_{KL}\big(q_{\phi}(z_t|y_t,h_t)||p_{\theta}(z_t)\big) + D_{KL}\big(q_{\phi}(y_t|h_t)||p_{\theta}(y_t)\big)\big\}\bigg\} + \alpha\mathcal{F}(y_{1:T}, l) \quad (10)$$

Another benefit of such a hierarchical model, over the basic one, is that it increases the complexity of the latent representation—$z_t \sim \{q_{\phi}(z_t) = \int q_{\phi}(z_t, y_t)dy_t\}$ is multi-modal even though $y_t$ and $z_t|y_t$ are assumed to be Gaussian—while keeping inference relatively simple since each of the latent distributions is conditionally a single Gaussian. In this sense the motivation is similar to that of existing work by Rezende & Mohamed (2015) and Maaløe et al. (2016). A final motivation, similarly to Sønderby et al. (2016); Zhao et al. (2017), is that this 2-layer model can directl represent a hierarchical feature space. Unlike prior work, however, a central goal here is to keep the task-specific latent variable uni-modal so that its mean can be used as the learned representation. The multi-view recurrent model of Equation 9 can also be extended with hierarchical latent variables, as shown in Figure 2(c). The resulting ELBO and derivation is given in the supplementary material.

### 3.3 PRIOR UPDATING

A simple prior $p(z) \equiv \mathcal{N}(0, I)$ is most common in vanilla VAEs and other variational approaches (Higgins et al., 2016; Alemi et al., 2016). Some recent work (Serban et al., 2017; Tomczak & Welling, 2017; Goyal et al., 2017b) has considered more complex prior distributions in order to better match the true distribution of the assumed latent space. In models of sequence data, existing variational approaches use priors on $z_t$ that are conditioned on various parts of the input and latent variables in other frames (Goyal et al., 2017a; Fraccaro et al., 2016; Chung et al., 2015b).

Motivated by the view of the prior as a regularizer, we propose an approach where the encoder/decoder ($\phi$ and $\theta$) and the prior are alternately updated during training. In particular, we set the prior in a given iteration $i$ to be the posterior from the previous iteration: $p_{\pi}^{(i)}(z_{1:T}) := q_{\phi^{(i)}}(z_{1:T}|x_{1:T})$, where $p_{\pi}^{(0)}(z_{1:T}) \equiv \mathcal{N}^T(0, I)$ and $\phi^{(i)}$, $\theta^{(i)}$ are the solutions maximizing the ELBO with respect to $p^{(i-1)}(z_{1:T})$. It is straightforward to show that

$$\max_{\phi,\theta}\bigg\{E_{q_{\phi}(z_{1:T}|x_{1:T})}\big[\log(p_{\theta}(x_{1:T}|z_{1:T}))\big] - \beta D_{KL}\big(q_{\phi}(z_{1:T}|x_{1:T})||p_{\pi}^{(i)}(z_{1:T})\big)\bigg\}$$

$$\leq \max_{\phi,\theta}\bigg\{E_{q_{\phi}(z_{1:T}|x_{1:T})}\big[\log(p_{\theta}(x_{1:T}|z_{1:T}))\big] - \beta D_{KL}\big(q_{\phi}(z_{1:T}|x_{1:T})||p_{\pi}^{(i+1)}(z_{1:T})\big)\bigg\} (11)$$

The intuition of this approach is that the KL regularization term should have a large effect at the beginning of training, and the effect should progressively diminish during training. More motivations and illustrations are given in the supplementary material.

## 4 EXPERIMENTS

The main goal of our work is to learn good representations for downstream tasks. We compare models on several tasks: (1) the CoNNL 2003 named entity recognition task Tjong Kim Sang & De Meulder (2003) and (2) the CoNNL 2000 text chunking task Tjong Kim Sang & Buchholz (2000); (3) phonetic recognition on the TIMIT speech corpus Zue et al. (1990) and the University of Wisconsin X-ray Microbeam (XRMB) acoustic-articulatory data set Westbury et al. (1990) (the latter used for multi-view representation learning), (4) character-level speech recognition on the Wall Street Journal (WSJ) data set Paul & Baker (1992). The first of these tasks was also studied by Chen et al. (2018), but only in a semi-supervised setting. In order to save space, detailed information on all data sets and hyperparameter tuning is included in the supplementary material.

In the single-view setting, we compare the following models: 1) Variational model with forward stochastic recurrent connections (StocCon, Figure 1(a)), 2) StocCon with prior updating (StocCon+P), 3) StocCon with both forward and backward (with the dependence path $z_T \rightarrow z_{T-1} \rightarrow \cdots$)

Table 1: F1 score of NER on CoNLL 2003 and Chunking on CoNLL 2000. "Baseline" = Two layer bidirectional GRU recognizer without any representation learning loss.

| Model | NER DEV | NER TEST | Chunking DEV | Chunking TEST |
|---|---|---|---|---|
| 1.Baseline | 93.3 | 89.3 | 94.1 | 93.1 |
| 2.StocCon | 92.4 | - | 92.8 | - |
| 3.StocCon+P | 93.2 | - | 94.0 | - |
| 4.RecRep+P | 93.6 | - | 94.1 | - |
| 5.RecRep+H+P | 93.7 | 89.8 | 94.5 | 93.7 |

Table 2: TIMIT phonetic error rates (%). "Baseline" = CTC recognizer without representation learning loss.

| Model | DEV | TEST |
|---|---|---|
| 1.Baseline | 17.5 | 19.4 |
| 2.StocCon | 17.6 | - |
| 3.StocCon+P | 17.5 | - |
| 4.StocCon+B+P | 16.9 | - |
| 5.RecRep ($\delta = 0$) | 18.0 | - |
| 6.RecRep+H | 17.4 | - |
| 7.RecRep+P | 17.2 | - |
| 8.RecRep+H+P | 16.7 | 19.0 |

Table 3: WSJ character error rates (%). "Baseline" = CTC recognizer without any representation learning loss. "#lab./unlab." refers to the number of labeled/unlabeled utterances used for training.

| Model | #lab. | /#unlab. | DEV | TEST |
|---|---|---|---|---|
| 1.Baseline1 | $5K$ | 0 | 25.3 | 22.8 |
| 3.StocCon+P | $5K$ | 0 | 25.5 | - |
| 4.RecRep+P | $5K$ | 0 | 24.8 | - |
| 6.RecRep+H+P | $5K$ | 0 | 23.1 | - |
| 5.RecRep+P | $5K$ | $18K$ | 22.2 | - |
| 7.RecRep+H+P | $5K$ | $18K$ | 20.7 | 18.1 |

stochastic recurrent layers (StocCon+B+P), , 4) Our model, variational recurrent model for representation learning, RecRep for short, 5) RecRep with prior updating (RecRep+P), 6) RecRep with hierarchical latent variables (RecRep+H) and 7) RecRep with both hierarchical latent variables and prior updating (RecRep+H+P).

In the multi-view setting, the models are our recurrent extensions of variational canonical correlation analysis (VCCAP) Wang et al. (2016): 1) Recurrent VCCAP with stochastic recurrent connections with prior updating (StocConVCCAP+P), 2) Our model, recurrent VCCAP without stochastic recurrent connections (RecVCCAP), 3) RecVCCAP with prior updating (RecVCCAP+P), 4) RecVC-CAP with latent hierarchy (RecVCCAP+H) and 5) RecVCCAP with hierarchy and prior updating (RecVCCA+H+P, Figure 2(b)).

### 4.1 OVERVIEW OF RESULTS

The results below follow the same general pattern across tasks. Models with (unidirectional) stochastic recurrent connections (StocCon) struggle to produce representations that improve over baselines, even when enhanced with prior updating. Bidirectional stochastic recurrent models (StocCon+B) improves performance, but at the cost of even more complex inference (for this reason we only attempted this model on one task, Table 2). Our RecRep models, without stochastic recurrent connections, always improve over the baselines at least when using prior updating and/or hierarchical latent variables. In some experiments we set $\delta = 0$ in the stochastic generation model (i.e., generating only $x_t$ from $z_t$); this always significantly hurts performance, showing the benefit of stochastic generation. Depending on the task, either prior updating or hierarchical latent variables is particularly useful. In all cases the best results are produced by RecRep+H+P models.

### 4.2 SUPERVISED SETTING: CONLL 2003 NAMED ENTITY RECOGNITION AND CONLL 2000 CHUNKING

For named entity recognition (NER) and text chunking (Table 1) we follow the setting of Peters et al. (2017). The input is a pretrained word embedding (GloVe 100-dimensional embedding Pennington et al. (2014)) concatenated with the output of a character RNN embedding. The baseline prediction model is a 2-layer bidirectional GRU RNN Cho et al. (2014); Chung et al. (2015a). . The decoder distribution $p_\theta(x|z)$ is a softmax rather than Gaussian, as in Miao et al. (2016). A more detailed generation model description is in the supplementary material.

### 4.3 SUPERVISED SETTING: TIMIT PHONETIC RECOGNITION

For phonetic recognition on the TIMIT data set, we use the same data processing and train/dev/test split as in Tang et al. (2017).[1] The baseline recognizer is a 3-layer stacked bidirectional LSTM Hochreiter & Schmidhuber (1997) network with pyramidal subsampling as in Chan et al. (2016); Lu et al. (2016), where the output of each layer (except the topmost layer) is reduced in length by a factor of two. The supervised phonetic recognition loss is the connectionist temporal classification (CTC) loss Graves et al. (2013). The representation learning loss (ELBO) is applied to the second layer. For all speech models, the decoder distribution $p_\theta(x|z)$ is modeled as a spherical Gaussian, with diagonal covariance matrix $\sigma^2 \mathbf{I}$, where $\sigma$ is a hyperparameter.

### 4.4 SEMI-SUPERVISED SETTING: WSJ CHARACTER-LEVEL SPEECH RECOGNITION

For character-level speech recognition on the Wall Stree Journal data set (WSJ), we use the standard dev (503 utterances) and test (330 utterances) sets. For semi-supervised experiments, we use about 5K utterances out of the full WSJ training set as labeled data and an additional subset of 16K utterances as unlabeled data. The recognizers are two-layer bidirectional GRU (BiGRU) networks trained with character-level CTC loss as in Miao et al. (2015). The representation learning loss (ELBO) is applied to the first layer BiGRU output. The results (Table 3) show that our models can improve performance both in the fully supervised setting (with a small amount of labeled data) and further in the semi-supervised setting (with some additional unlabeled data).

### 4.5 MULTI-VIEW SETTING: XRMB PHONETIC RECOGNITION

We use the same setup as in prior work on XRMB Tang et al. (2018): We use the acoustic+articulatory data from 35 speakers for representation learning and the acoustic-only data from 12 speakers for phonetic recognition experiments in a 6-fold setup (Table 4). The recurrent models are 2-layer bidirectional LSTMs with 256 units per layer. We also include a result from prior work using basic VCCAP with non-recurrent fully connected feedforward networks, which operate on large overlapping fixed-size segments of the input Tang et al. (2018). Interestingly, here all of the representation learning approaches improve significantly over baseline, and the non-recurrent feedforward models perform quite well, although the best performance is again produced by the proposed RecVCCAP models with stochastic generation. In anecdotal experiments, we find that we can further improve phonetic error rates to $< 7\%$ with either feedforward or RecVCCAP models by increasing their complexity (e.g., by further increasing the window size in the feedforward models); however, the memory and computation time requirements of feedforward models quickly become prohibitive while the recurrent models remain fairly efficient.

Table 4: XRMB phonetic error rates (%) with multi-view representation learning models.

| Model | 6-fold PER |
|---|---|
| 1.Baseline | 12.9 |
| 2.Baseline (4-layer) | 11.1 |
| 3.StocConVCCAP+P | 9.9 |
| 4.Feed-Forward  Tang et al. (2018) | 9.4 |
| 5.RecVCCAP ($\delta = 0$) | 9.7 |
| 6.RecVCCAP | 9.2 |
| 7.RecVCCAP+P | 8.9 |
| 8.RecVCCAP+H | 8.8 |
| 9.RecVCCAP+H+P | 7.3 |

### 4.6 VISUALIZATION OF LEARNED REPRESENTATIONS

We visualize the features we learned on XRMB in the multi-view setting via t-SNE embeddings Maaten & Hinton (2008). We compare three types of features in Figure 3, computed on 443 randomly selected speech frames each corresponding to one of the 6 vowels [aa,ae,uw,ih,eh,ow]

---

[1]We do not, however, use a second validation set as in Tang et al. (2017).

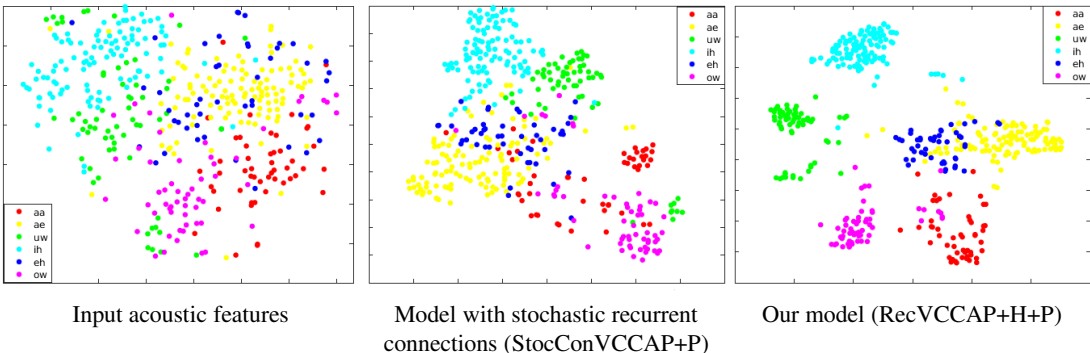

| Input acoustic features | Model with stochastic recurrent connections (StocConVCCAP+P) | Our model (RecVCCAP+H+P) |

Figure 3: t-SNE plots of several XRMB speech representations, corresponding to rows 1, 3, and 9 of Table 4.

from the 12 test speakers. Qualitatively, these visualizations show that representations learned with the proposed models form tighter label-specific clusters.

## 5 CONCLUSION

We have proposed an approach for learning sequence representations of sequence inputs via variational recurrent neural models. Unlike most prior work, our approach focuses on the quality of the learned representation for a variety of downstream tasks rather than on generation quality, and the representations are learned in a multitask setting along with a supervised, semi-supervised, or weakly supervised multi-view loss. We use simple, efficient inference networks where the mean of the latent variables naturally serves as a representation, and we introduce stochastic generation as an approach for modeling context without a complex inference model. We have studied the proposed models in the context of several speech and language tasks. We find that the proposed recurrent representations consistently improve task performance, and that hierarchical latent variables and prior updating are useful in a variety of settings.

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

# SUPPLEMENTARY MATERIAL: VARIATIONAL RECURRENT MODELS FOR REPRESENTATION LEARNING

**Anonymous authors**

## 1 DERIVATION FOR STOCHASTIC GENERATION

As mentioned in the main text, we are not calculating the full generative model shown in Figure 1(d) of the main paper. Instead, we use the process referred to as stochastic generation. Here, we illustrate the connection between using stochastic generation for training and directly evaluating the multi-modal generative model corresponding to Figure 1(d) in the main paper.

Assume we have $N$ sequences in our training set, and assume each sequence has been visited (via stochastic generation) $K$ times. We denote the $t^{th}$ latent variable of sequence $x_{1:T}$ as $z_t$. Suppose $z_t$ has tried to reconstruct $x_k$ for $M_{t,k}$ times. So on average, by using stochastic generation in training, the log-likelihood computed for the sequence $x_{1:T}$ given the posterior is as follows:

$$\Sigma_{t=1}^T \left\{ \mathbb{E}_{q_\phi(z_t|h_t)} \left[ \Sigma_{k=1}^T \left\{ \frac{M_{t,k}}{K_i} \log p_\theta(x_k|z_t) \right\} \right] \right\} \tag{1}$$

with $\mathbf{E}[\frac{M_{t,k}}{K}] = \alpha_{\delta,T}^{(t,k)}$ given the definition of stochastic generation.

If we directly evaluate the generation distribution

$$\begin{aligned} p_\theta(x_{1:T}|z_{1:T}) &= \Pi_{t=1}^T p_\theta(x_t|z_{1:T}) \\ &= \Pi_{t=1}^T \Pi_{k=1}^T p_\theta(x_k|z_t)^{\alpha_{\delta,T}^{(t,k)}} \end{aligned} \tag{2}$$

then we can derive the ELBO as follows:

$$\begin{aligned} \log p_\theta(x_{1:T}) &\geq \mathbb{E}_{q_\phi(z_{1:T}|h_{1:T})} \left[ \log p_\theta(x_{1:T}|z_{1:T}) \right] - D_{KL}\left( q_\phi(z_{1:T}|h_{1:T}) || p(z_{1:T}) \right) \\ &= \mathbb{E}_{q_\phi(z_{1:T}|h_{1:T})} \left[ \Sigma_{t=1}^T \left\{ \Sigma_{k=1}^T \left\{ \alpha_{\delta,T}^{(k,t)} \log p_\theta(x_t|z_k) \right\} \right\} \right] - D_{KL}(q_\phi(z_{1:T}|h_{1:T}) || p(z_{1:T})) \\ &= \mathbb{E}_{q_\phi(z_{1:T}|h_{1:T})} \left[ \Sigma_{t=1}^T \left\{ \Sigma_{k=1}^T \left\{ \alpha_{\delta,T}^{(t,k)} \log p_\theta(x_k|z_t) \right\} \right\} \right] - D_{KL}(q_\phi(z_{1:T}|h_{1:T}) || p(z_{1:T})) \\ &= \Sigma_{t=1}^T \left\{ \mathbb{E}_{q_\phi(z_t|h_t)} \left[ \Sigma_{k=1}^T \left\{ \alpha_{\delta,T}^{(t,k)} \log p_\theta(x_k|z_t) \right\} \right] - D_{KL}(q_\phi(z_t|h_t) || p(z_t)) \right\} \end{aligned} \tag{3} \tag{4}$$

The log-likelihood of $x_{1:T}$ given the posterior is then as follows:

$$\Sigma_{t=1}^T \left\{ \mathbb{E}_{q_\phi(z_t|h_t)} \left[ \Sigma_{k=1}^T \left\{ \alpha_{\delta,T}^{(t,k)} \log p_\theta(x_k|z_t) \right\} \right] \right. \tag{5}$$

According to Equation 1, in expectation, the conditional log-likelihood computed using stochastic generation is also equal to Equation 5. This equality shows the connection between stochastic generation and the full generation process.

## 2 PROOF AND ILLUSTRATION OF PRIOR UPDATING

Given the prior $p_\pi(z_{1:T})$ parameterized by $\pi$, we have the ELBO

$$
\begin{aligned}
\log p_{\theta,\pi}(x_{1:T}) &= \log \frac{p_\theta(x_{1:T}|z_{1:T})p_\pi(z_{1:T})q_\phi(z_{1:T}|x_{1:T})}{q_\phi(z_{1:T}|x_{1:T})p_\theta(z_{1:T}|x_{1:T})} \\
&= \mathbb{E}_{q_\phi(z_{1:T}|x_{1:T})} \log \frac{p_\theta(x_{1:T}|z_{1:T})p_\pi(z_{1:T})q_\phi(z_{1:T}|x_{1:T})}{q_\phi(z_{1:T}|x_{1:T})p_\theta(z_{1:T}|x_{1:T})} \\
&\geq \mathbb{E}_{q_\phi(z_{1:T}|x_{1:T})} \log p_\theta(x_{1:T}|z_{1:T}) - D_{KL}\big(q_\phi(z_{1:T}|x_{1:T})||p_\pi(z_{1:T})\big) \quad (6)
\end{aligned}
$$

As described in the paper, when using prior updating, we simply set $p_\pi^{(i)}(z_{1:T}) := q_{\phi^{(i)}}(z_{1:T}|x_{1:T})$. Here, $p_\pi^{(i)}(z_{1:T})$ is the prior after $i$ updates, where $p_\pi^{(0)}(z_{1:T}) \equiv \mathcal{N}^T(0, I)$; $\phi^{(i)}$ and $\theta^{(i)}$ are the solutions maximizing the lower bound with respect to $p^{(i-1)}(z_{1:T})$, so we have

$$
\begin{aligned}
&\max_{\phi,\theta} \left\{ E_{q_\phi(z_{1:T}|x_{1:T})} \big[ \log(p_\theta(x_{1:T}|z_{1:T})) \big] - \beta D_{KL}\big(q_\phi(z_{1:T}|x_{1:T})||p_\pi^{(i)}(z_{1:T})\big) \right\} \\
&= E_{q_{\phi^{(i+1)}}(z_{1:T}|x_{1:T})} \big[ \log(p_{\theta^{(i+1)}}(x_{1:T}|z_{1:T})) \big] - \beta D_{KL}\big(q_{\phi^{(i+1)}}(z_{1:T}|x_{1:T})||p_\pi^{(i)}(z_{1:T})\big) \\
&\leq E_{q_{\phi^{(i+1)}}(z_{1:T}|x_{1:T})} \big[ \log(p_{\theta^{(i+1)}}(x_{1:T}|z_{1:T})) \big] - \left\{ \beta D_{KL}\big(q_{\phi^{(i+1)}}(z_{1:T}|x_{1:T})||p_\pi^{(i+1)}(z_{1:T})\big\} = 0 \right. \\
&\leq \max_{\phi,\theta} \left\{ E_{q_\phi(z_{1:T}|x_{1:T})} \big[ \log(p_\theta(x_{1:T}|z_{1:T})) \big] - \beta D_{KL}\big(q_\phi(z_{1:T}|x_{1:T})||p_\pi^{(i+1)}(z_{1:T})\big) \right\} \quad (7)
\end{aligned}
$$

It is not difficult to show that the **equality** holds when and only when we have $q_{\phi^{(i+1)}}(z_{1:T}|x_{1:T}) = p_\pi^{(i)}(z_{1:T}) \equiv q_{\phi^{(i)}}(z_{1:T}|x_{1:T})$; that is, observing the evidence does not change our understanding of the latent space. An illustration of prior updating from a regularization perspective is given in Figure 1.

## 3 ELBO OF RECURRENT MULTI-VIEW (HIERARCHICAL) MODELS

We now derive the ELBO for RecVCCAP.

$$
\begin{aligned}
&\log(p_\theta(x_{1:T}, u_{1:T})) \\
&= E_{q_\phi(z_{1:T}, y_{1:T}, o_{1:T}|g_{1:T}, h_{1:T}, n_{1:T})} \Big[ \log \Big( \frac{p_\theta(x_{1:T}, u_{1:T}|z_{1:T}, y_{1:T}, o_{1:T})p_\theta(z_{1:T}, y_{1:T}, o_{1:T})}{p_\theta(z_{1:T}, y_{1:T}, o_{1:T}|x_{1:T}, u_{1:T})} \Big) \Big] \quad (8)
\end{aligned}
$$

Assume the prior distribution is given and factorizes as

$$
p_\theta(z_{1:T}, y_{1:T}, o_{1:T}) = p(z_{1:T}, y_{1:T}, o_{1:T}) = p(z_{1:T}) \times p(y_{1:T}) \times p(o_{1:T}) \quad (9)
$$

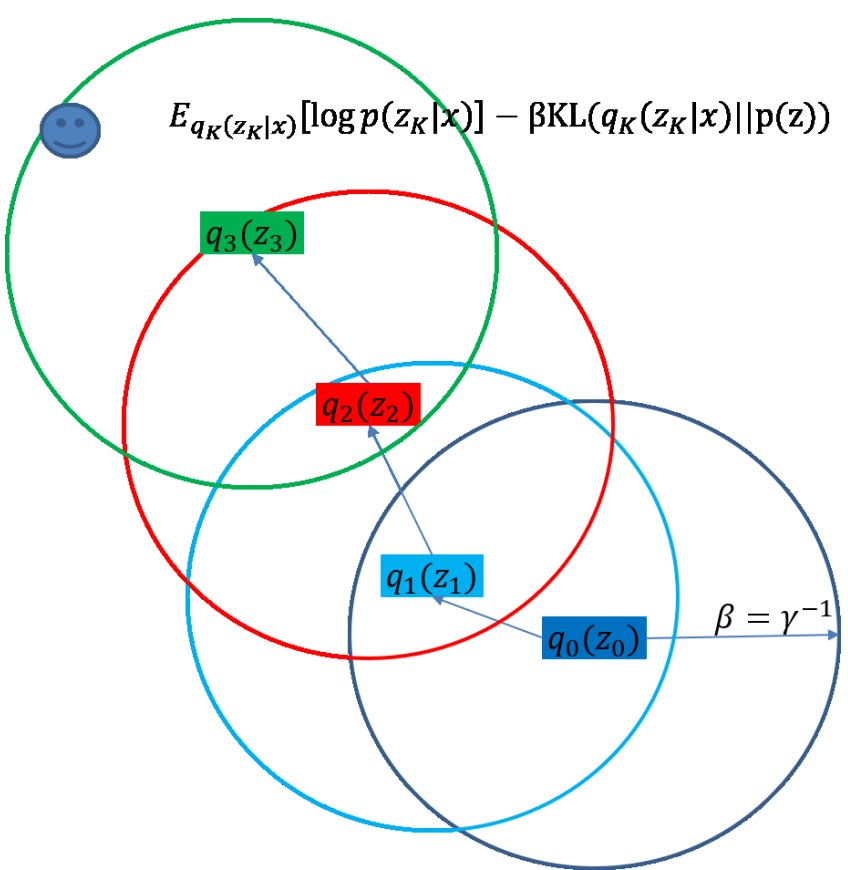

$$E_{q_K(z_K|x)}[\log p(z_K|x)] - \beta \text{KL}(q_K(z_K|x)||p(z))$$

Figure 1: Illustrative explanation of our prior updating strategy from a regularization perspective.

We then have

$$\log(p_\theta(x_{1:T}, u_{1:T}))$$

$$= E_{q_\phi(z_{1:T}, y_{1:T}, o_{1:T}|g_{1:T}, h_{1:T}, n_{1:T})}\Big[\log\big(\frac{p_\theta(x_{1:T}, u_{1:T}|z_{1:T}, y_{1:T}, o_{1:T})p(z_{1:T}, y_{1:T}, o_{1:T})q_\phi(z_{1:T}, y_{1:T}, o_{1:T}|g_{1:T}, h_{1:T}, n_1}{p_\theta(z_{1:T}, y_{1:T}, o_{1:T}|x_{1:T}, u_{1:T})q_\phi(z_{1:T}, y_{1:T}, o_{1:T}|g_{1:T}, h_{1:T}, n_{1:T})}\big)$$

$$= E_{q_\phi(z_{1:T}, y_{1:T}, o_{1:T}|g_{1:T}, h_{1:T}, n_{1:T})}\Big[\log\big(p_\theta(x_{1:T}, u_{1:T}|z_{1:T}, y_{1:T}, o_{1:T})\big)\Big]$$

$$+ E_{q_\phi(z_{1:T}, y_{1:T}, o_{1:T}|g_{1:T}, h_{1:T}, n_{1:T})}\Big[\log\big(\frac{q_\phi(z_{1:T}, y_{1:T}, o_{1:T}|g_{1:T}, h_{1:T}, n_{1:T})}{p_\theta(z_{1:T}, y_{1:T}, o_{1:T}|x_{1:T}, u_{1:T})}\big)\Big]$$

$$+ E_{q_\phi(z_{1:T}, y_{1:T}, o_{1:T}|g_{1:T}, h_{1:T}, n_{1:T})}\Big[\log\big(\frac{p(z_{1:T}, y_{1:T}, o_{1:T})}{q_\phi(z_{1:T}, y_{1:T}, o_{1:T}|g_{1:T}, h_{1:T}, n_{1:T})}\big)\Big]$$

$$= E_{q_\phi(z_{1:T}, y_{1:T}, o_{1:T}|g_{1:T}, h_{1:T}, n_{1:T})}\Big[\log\big(p_\theta(x_{1:T}, u_{1:T}|z_{1:T}, y_{1:T}, o_{1:T})\big)\Big]$$

$$+ D_{KL}(q_\phi(z_{1:T}, y_{1:T}, o_{1:T}|g_{1:T}, h_{1:T}, n_{1:T})||p_\theta(z_{1:T}, y_{1:T}, o_{1:T}|x_{1:T}, u_{1:T}))$$

$$- D_{KL}(q_\phi(z_{1:T}, y_{1:T}, o_{1:T}|g_{1:T}, h_{1:T}, n_{1:T})||p(z_{1:T}, y_{1:T}, o_{1:T}))$$

$$\geq E_{q_\phi(z_{1:T}, y_{1:T}, o_{1:T}|g_{1:T}, h_{1:T}, n_{1:T})}\Big[\log\big(p_\theta(x_{1:T}, u_{1:T}|z_{1:T}, y_{1:T}, o_{1:T})\big)\Big]$$

$$- D_{KL}\big(q_\phi(z_{1:T}, y_{1:T}, o_{1:T}|g_{1:T}, h_{1:T}, n_{1:T})||p(z_{1:T}, y_{1:T}, o_{1:T})\big)$$

$$= E_{q_\phi(z_{1:T}, y_{1:T}|g_{1:T}, h_{1:T})}\big[\log p_\theta(x_{1:T}|z_{1:T}, y_{1:T})\big] + E_{q_\phi(o_{1:T}, y_{1:T}|n_{1:T}, h_{1:T})}\big[\log p_\theta(u_{1:T}|o_{1:T}, y_{1:T})\big]$$

$$- D_{KL}\big(q_\phi(o_{1:T}|n_{1:T})||p(o_{1:T})\big) - D_{KL}\big(q_\phi(z_{1:T}|g_{1:T})||p(z_{1:T})\big) - D_{KL}\big(q_\phi(y_{1:T}|h_{1:T})||p(y_{1:T})\big)$$

$$= \Sigma_{t=1}^T\Big\{\mathbb{E}_{q_\phi(y_t|h_t)q_\phi(z_t|g_t)}\Big[\Sigma_{k=1}^T\big\{\alpha_{\delta,T}^{t,k}\log p_\theta(x_k|z_t)\big\} + \mathbb{E}_{q_\phi(y_t|h_t)q_\phi(o_t|n_t)}\Sigma_{k=1}^T\big\{\alpha_{\delta,T}^{t,k}\log p_\theta(u_k|o_t)\big\}\Big]\Big\}$$

$$- D_{KL}\big(q_\phi(o_{1:T}|n_{1:T})||p(o_{1:T})\big) - D_{KL}\big(q_\phi(z_{1:T}|g_{1:T})||p(z_{1:T})\big) - D_{KL}\big(q_\phi(y_{1:T}|h_{1:T})||p(y_{1:T})\big)$$

Similarly, we can derive the ELBO for the hierarchical version of RecVCCAP as follows:

$$
\begin{aligned}
\log p_\theta(x_{1:T}, u_{1:T}) \geq & \; E_{q_\phi(z_{1:T}, y_{1:T}|g_{1:T}, h_{1:T})}\big[\log_\theta(x_{1:T}|z_{1:T})\big] + E_{q_\phi(o_{1:T}, y_{1:T}|n_{1:T}, h_{1:T})}\big[\log_\theta(u_{1:T}|o_{1:T})\big] \\
- & \; D_{KL}\big(q_\phi(o_{1:T}|y_{1:T})||p(o_{1:T})\big) - D_{KL}\big(q_\phi(z_{1:T}|y_{1:T})||p(z_{1:T})\big) - D_{KL}\big(q_\phi(y_{1:T}|h_{1:T})||p(y_{1:T})\big) \\
= & \; \Sigma_{t=1}^{T}\Big\{\mathbb{E}_{q_\phi(y_t|h_t)q_\phi(z_t|y_t, g_t)}\Big[\Sigma_{k=1}^{T}\big\{\alpha_{\delta,T}^{t,k}\log p_\theta(x_k|z_t)\big\} + \mathbb{E}_{q_\phi(y_t|h_t)q_\phi(o_t|y_t, n_t)}\Sigma_{k=1}^{T}\big\{\alpha_{\delta,T}^{t,k}\log p_\theta(u_k|o_t)\big\}\Big]\Big\} \\
- & \; D_{KL}\big(q_\phi(o_{1:T}|y_{1:T})||p(o_{1:T})\big) - D_{KL}\big(q_\phi(z_{1:T}|y_{1:T})||p(z_{1:T})\big) - D_{KL}\big(q_\phi(y_{1:T}|h_{1:T})||p(y_{1:T})\big) \quad (11)
\end{aligned}
$$

## 4  DECODER DETAILS

For speech tasks we use a Gaussian distribution with diagonal covariance as the decoder distribution, that is

$$
\log p_\theta(x|z_t) \equiv \mathcal{N}(\mu|_{z_t}, \sigma^2 I) \tag{12}
$$

where $\sigma$ is tuned.

For NLP tasks we use a softmax decoder as described in Equation (6) and (7) of Miao et al. (2016). In our tasks (NER and chunking), the input is a character RNN output concatenated with a pre-trained word embedding, and the generation network tries to reconstruct the word corresponding to the given embedding by matching the reconstructed embedding with the (pre-trained) embedding of the target word.

## 5  DATA SETS

In this section, we give more details for the five data sets we used in the paper: TIMIT, XRMB, WSJ, CoNLL2003 and CoNLL2000.

For TIMIT, there are 3696, 400 and 192 train/dev/test utterances respectively. The per frame input are speaker-normalized 40-dimensional log filter bank features (without energy) and their first and second derivatives. 39 phone labels are used.

For XRMB, there are in total 47 speakers and roughly 2300 acoustic utterances. 35 speakers (roughly 1700 utterances), together with their paired articulatory measurements, are used to train multi-view models. The remaining 12 speakers are used to test the quality of the learned representations by testing on phonetic recognition in a $K$-fold experimental setup. We use 6-fold validation here, where the 12 speakers (separate from the 35 speakers used for multi-view representation learning) are separated into 6 groups, and each time we pick one group as development set, one group as test set, and the remaining 4 groups (8 speakers) as phonetic recognizer training set. For XRMB, the input features are 13-dimensional MFCCs, also with first and second derivatives, making the feature vectors 39-dimensional.

For WSJ, we start with the standard split, with 37416 train, 503 dev, and 330 test utterances. However, in this paper we don't use the full training set. For WSJ we again use log filter bank features, but with energy, and also their first and second derivatives. So the per-frame input feature vectors are 123-dimensional. For the purpose of semi-supervised learning, we split the training into 24 parts, and use 3 of them (roughly $5K$ utterances) as labeled data and treat the other 21 splits as unlabeled data.

For the CoNLL2003 and CoNLL2000 tasks, we refer the reader to Peters et al. (2017), specifically the second and fourth paragraphs of the experimental section.

## 6  HYPERPARAMETER TUNING AND TRAINING

For all speech models, the decoder distribution $p_\theta(x|z)$ is a spherical Gaussian, with covariance matrix $\sigma^2 \mathbf{I}$. $\sigma$ is a hyperparameter selected among $\{0.1, 1.0, 10.0\}$. The KL term weight $\beta$ is selected among $\{0.1, 1.0, 10.0\}$, $\alpha$ is selected among $\{0.3, 0.5, 0.7, 0.9, 0.99\}$, and $\sqrt{\delta}$ is selected among $\{0, 0.05, 0.1, 0.5, 1, 2, 5, 10, 20\}$. The tuning strategy applied to all of the tasks in this work.

For learning, we typically use Adam Kingma & Ba (2015), with initial learning rate chosen from $\{0.001, 0.0005, 0.0001\}$. For TIMIT, we also use stochastic gradient descent, following the optimization strategy described in Tang et al. (2017).

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
