# OpenReview forum: "Variational recurrent models for representation learning"
_ICLR.cc/2019/Conference_

### Official Review · AnonReviewer2 · 2018-11-02
**Incremental contribution of variational recurrent models; big volume of extensions of the proposed method and experiments**

**Rating:** 5
**Confidence:** 3

**Review:**

This paper proposes a new variational recurrent model for learning sequences. Comparing to existing work, instead of having latent variables that are dependent on the neighbors, this paper proposes to use independent latent variables with observations that are generated from multiple latent variables.
The paper further combined the proposed method with multiple existing ideas, such as the shared/prviate representation from VAE-CCAE, adding the hierarchical structure, and prior updating.

Pros:
The proposed method seems technical correct and reasonable.
There are many extensions which are potentially useful for many applications
There are many experimental results showing promising performance.

Cons:
The framework is very incremental. It is novel but limited.
The paper claim that the main point to use the simpler variations distribution is to speed up the inference. But no speed comparisons are shown in the experiments section.
The evaluation shows that prior updating (one extension) seems contributes to the biggest performance gain, not the main proposed method.

---

> ### Author Response · Authors · 2018-11-23
> **On clarifying contribution and speed comparison**
>
> Thank you for pointing out the missing speed comparison.  RecRep is roughly twice faster in our implementation than StocCon when using batch size 4.  We will include this in a revision.  Regarding the degree of novelty, our main contribution is a practical approach to representation learning for sequences that improves performance on multiple downstream tasks.  While prior work has largely focused on measuring the quality of recurrent models for generation, we focus on making them useful for representation learning.

---

### Official Review · AnonReviewer3 · 2018-11-04
**Method that tries to be a feature extractor and a generative model at the same time.**

**Rating:** 3
**Confidence:** 5

**Review:**

(best read in typora)

The authors claim to propose a family of methods and generative models that are suited better for downstream tasks than previously proposed approaches.

## Major points

It feels as if the proposed method tries to be many things. First, it is used for finding unsupervised representations down stream. Then, it still tries to be a generative model "of sorts", which is the reason for the use of variational inference in the first place. Additionally, the approximate posterior necessary to evaluate the ELBO is simultaneously used as a feature extractor.

The resulting issues are:

  - A "bad" variational posterior is used because it is unclear how to get vectorial features otherwise.
  - An adhoc likelihood function is used, which is not sufficiently well explored theoretically in the paper.  Specifically,
      - Stochastic generation is claimed to be "more complex than simple Gaussian"; the burden of proof is on the authors, as Gaussian density is closed under multiplication.
      - It appears to be a Monte Carlo approximation to sth that is computable in closed form.
      - It is not clear if that MC approximation is normalised and if the normalisation is the same at each optimisation step. Does this bias optimisation? What happens to the KL penalty weight?
  - The ELBO change (prior updating) seems to make the claim that we still have a generative model (as written in the intro) invalid. My intuition is that the KL penalty vanishes for small step rates of the optimiser, reducing the model to that of a noisy auto encoder.


## Summary

The authors want to evaluate variational sequence models for feature extraction for downstream tasks. But why? What is the use of a generative inspired algorithm, when necessary ingredients are discarded? Both goals appear to be at conflict and I am not convinced that the variational ingredient is necessary.

I do not cover the experimental section since the method itself has issues so severe that I don't consider it relevant.


## Minor points

- Notation $\mu_{\phi_t}$ gives the impression that $\phi$ is time dependent.
- Equations (9) and (11) are formatted badly.
- The approximate posterior used was used first in (Bayer & Osendorfer, "Learning stochastic recurrent networks", 2014) not (Chen 2018).
- Diagrams follow GM notation only half-heartedly.

---

> ### Author Response · Authors · 2018-11-23
> **Clarification on stochastic generation and more explanations**
>
>
> Q1: It feels as if the proposed method tries to be many things. First, it is used for finding unsupervised representations down stream. Then, it still tries to be a generative model "of sorts", which is the reason for the use of variational inference in the first place. Additionally, the approximate posterior necessary to evaluate the ELBO is simultaneously used as a feature extractor.
> Ans: Our only goal is representation learning. VAEs have been successful for representation learning for non-sequential data (e.g. higgins2017beta, alemi2016deep), and we aim to design recurrent counterparts that are similarly useful for representation learning for sequences. The generative model is useful as a tool for learning representations (as has been found in prior work on representation learning as well), but we are not trying to learn a model that generates well. The use of the approximate posterior in a VAE as a feature extractor is fairly commonplace, e.g. kingma2014semi, maaloe2016auxiliary, zhou2017morphological. Indeed, our results show that we get significant improvements on multiple downstream tasks, which to our knowledge has not been done before with recurrent sequence models for representation learning.
>
> Q2: A “bad" variational posterior is used because it is unclear how to get vectorial features otherwise.
> The “badness'' of the posterior is perhaps in the eye of the beholder :)  Our posterior has multiple advantages, both the ability to easily get features and avoidance of complex sampling procedures.
>
> Q3 on “stochastic generation”
> Ans:
> a.Agreed, the approach could be explored further theoretically.
> b. The likelihood for each x_k is in fact a mixture of T Gaussians when each p(x_k|z_t) is Gaussian; see detailed derivation in second section here:
> https://drive.google.com/file/d/1FomO05-wiLVFm4W5zNrpZW04_Gj0jwiM/view?usp=sharing
> There is a typo in Equation (6) in the current version, which should be revised to \sum_{t=1}^T\{ \mathbb{E}_{q_{\phi}(z_t|h_t)}\big[ \log \sum_{k=1}^T\alpha_{\delta,T}^{t,k} p_{\theta}(x_k|z_t) \big] \}.
> c. In general, this is computable in closed form but in O(T^2) time for a length T sequence.  Our approximation allows us to compute it in O(T)$ time for any distribution.
> d.  Yes, it is normalized. See our reply to bullet point (b) and the anonymized link.
>
> Q4 on ELBO change and prior updating
> Ans: That is our intuition as well.  We are not making a claim about the generative model, but taking an optimization view of our approach.  Viewed this way, a VAE is indeed a “noisy autoencoder'' with an additional KL regularization term.  Upon updating the prior, we search for a better model in the “vicinity'' of the previously learned model, using the previously learned networks as a warm start.  This is in spirit similar to annealing the weight of the KL term, but is data-dependent and requires no tuning of weight parameters.  In a small comparison experiment, we find that KL annealing comes close to our performance improvement, but at the cost of tuning multiple weight parameters.  We will include these results in a revision.
>
> Q5 in summary section
> Ans: We agree there is no special reason to favor variational or generative model for representation learning, other than that they work well and provide an intuitive way to reason about regularization.  We note that generative models (e.g., HMMs) have been frequently used for non-generative tasks.  From an optimization point of view, as mentioned above VAEs are “noisy autoencoders'' with a KL regularization term, and have been more successful than other autoencoder-type models for non-sequence tasks.  This motivates the application to sequence tasks.

---

### Official Review · AnonReviewer1 · 2018-11-05
**Needs stronger motivation, better analysis would improve the paper**

**Rating:** 5
**Confidence:** 3

**Review:**

This is largely an experimental paper, proposing and evaluating various modifications of variational recurrent models towards obtaining sequence data representations that are effective in downstream tasks. The highlighted contribution is a "stochastic generation" training procedure in which the training objective evaluates the reconstruction of output sequence elements from individual latent variables independently. The main claim is that the resulting model, augmented with prior updating and/or hierarchical latent variables, improves results w.r.t. the baselines.

My main concern is that the various choices are not motivated well, e.g. with examples or detailed descriptions of the issues addressed and that the resulting implications are not discussed in detail (see detailed comments below). This could perhaps be alleviated during the rebuttal discussion.

Empirically, when used in conjunction with prior updating and/or hierarchical latent variables, the proposed "stochastic generation" approach improves upon the baselines, but not when used in isolation. This is OK, but it weakens the contribution since it's more unclear what the exact advantage "stochastic generation" is, how it takes advantage of prior updating, and so on. Could you maybe discuss this in the rebuttal? The fact that not all model variants considered are evaluated on all settings also contributes to this problem (again, see below).

General questions:
- "dependence of observations at each time step on all latent variables": Unfortunately, this means that the complexity of evaluating the model during training is O(n^2), where n is the sequence size, rather than linear in the standard case. Is that correct? I think this is what is alluded to on the top on page 4. Could you discuss this trade-off?
- regarding section 2.1.: Multi-modal marginal probabilities are also used due their increased modeling power, and this again seems like a potential limitation of the proposed approach w.r.t. the baseline, and is not discussed.
- "the mean of z_t may have very small probability and thus may not be a good choice": I think this statement requires more context. The mean of z_t can have low probability in both cases (e.g. if the posterior has a high variance). Are you suggesting that the low probability issue is exacerbated by to the sampling of previous z_{t-1}? Or are you comparing to the case where the mean z_{t-1} is used instead of sampling as well?

Stochastic generation:
- While I understand where it's coming from, the term "stochastic generation" is somewhat misleading, since stochasticity is already present in the generation process for VAEs;
- Stochastic generation is introduced as a way to approximate the generation process. However, when it's introduced, it's not clear what the generation process that needs to be approximated is. Introducing the model in eq. (6-7), motivating its use and then showing how it is obtained through stochastic generation second would improve the clarity of the paper.
- Related to the point above, the implications of using the model in eq. (6-7) are not discussed. The graphical model in Figure 1 suggests that x_k depends jointly on all the (z_t)_{t=1 ... sequence_size}. Instead, in eq. (6-7), each x_k is generated independently from each z_t (for t = 1 ... T, and k sampled from a distribution which depends on t). In particular, if I understand this correctly, the distribution p(x_k | z) = p(x_k | z_1 \dots z_T) factorizes as p(x_k | z_1) p(x_k | z_2) ... p(x_k | z_T). Could you motivate this choice and its expected effect? It seems to me that this encourages each z_t to capture all the information needed to reconstruct each x_k in the corresponding window.

Experimental results:
- Table 2: I think this table since it includes most models, but it still misses RecRep (without delta = 0) and StocCon. Could you confirm whether StocCon vs. RecRep have the same setting except the use of recurrent stochastic connections in StocCon vs. using eq. (4) in RecRep with window size 1?
- In Table 4, the difference between line 5 and line 6 is interesting and I wish it was discussed more, maybe used in the visualization experiment to show how/why "stochastic generation" with a larger window improves performance.
- Figure 3, could it be that the use of hierarchical latent variables (H) accounts for the visual difference? Is a difference still observed when comparing lines 3 and 7 in Table 4, whose settings seem more comparable?
-

Minor issues:
- the lack of parenthesis around citations makes the text hard to follow at times (maybe use \citep whenever the citation mixes with the text?);
- typo: "for use in a downstream tasks"
- typo: "with graphical model as described" => "with the/a graphical model as described"

---

> ### Author Response · Authors · 2018-11-23
> **Explanations on motivations, details and visualization**
>
>
> Main concern on motivation
> Ans: Thanks for pointing this out.  We agree the motivations could be clearer and will revise the paper to remedy this.
> All of the choices are intended to better handle the sequence structure of the input, in the context of representation learning, rather than generation. Specifically, the main motivations are:
> 1. Stochastic Generation: With stochastic generation we encourage the learned features to reconstruct not only the current frame, but also nearby frames, therefore incorporating more context information. This approach is intended to be an easier way to model context than via stochastic recurrent connections among the latent variables; it is both more efficient and more natural to use the mean of the latent representation when we have no stochastic recurrent connections (see more on this below).
> 2. Hierarchical Latent Variables: There are two motivations.  First, the two latent variables each focus on different tasks (recognition and reconstruction in our case), so the latent variable corresponding to the downstream task need not store unnecessary information needed only for reconstruction.
> Second, the use of hierarchical latent variables makes the inference model more powerful since z_t now depends on the sample of y_t, while still allowing efficient computation of the mean of y_t and faster test-time performance since we only use the mean of q(y_t) as features.
> 3. Prior Updating: It is natural to use a structured, time-dependent prior distribution in dealing with sequences, and this is enabled by prior updating in this work. Upon updating the prior, we search for a better model in the "``vicinity'' of the previously learned model, using the previously learned networks as a warm start.  This is in spirit similar to annealing the weight of the KL term as in other work, but is data-dependent and requires no tuning of weight parameters.
>
> Main concern on experiments
> Ans:  We have performed a more detailed ablation on the TIMIT development set. Both stochastic generation and prior updating reduce the error rate by 0.2% (absolute), but hierarchical latent variables do not reduce the error rate. The combination of all three reduces the error rate by 0.6%.  On other tasks (e.g. one of the reported NLP tasks), hierarchical latent variables do help.  It would be more satisfying if there was a universal model that worked best on all tasks, but for the time being it seems there are some task-specific details favoring certain techniques.
>
> General Q1
> Ans: The cost is not actually O(n^2), as we do not use the full bipartite graphical model shown in Figure 1(d).  Specifically, for each latent variable, in each epoch we only generate a single time step of the observation sequence, with higher probability for nearby time steps, which has cost O(n).
>
> General Q2
> Ans: We are afraid we do not completely understand this question.  Why is increased modeling power a limitation?  If you could clarify a bit that would be great.
>
> General Q3
> Ans: Yes, the mean of z_t can have arbitrarily low probability/density in any distribution, but in a unimodal distribution, the mean is likelier than any other value, making it a natural choice.  If the distribution is multimodal, it is not clear what single value to consider as the learned representation.  (This is on top of the fact that we may also need many sample paths z_1 \rightarrow z_2 \rightarrow ... \rightarrow z_{t-1} \rightarrow z_t and complex sampling to estimate the distribution of z_t).
>
> Stochastic generation Q1
> Ans: Thanks for pointing out this. Yes, "``stochasticity" naturally exists in VAEs. We are using this term to refer to a "``stochastic target", that is, in generating x_{?} from z_t.
>
> Stochastic generation Q2
> Ans: Nice suggestion! We will reorganize.
>
> Stochastic generation Q3
> Ans: Your intuition is correct, that the goal is for each $z_t$ to capture more information about the temporal context.
>
> Question on experiments
> Ans: Yes, the settings are the same.  We will add the additional ablation studies mentioned above.
> We have visualized all of the variants in Table 4. Please see https://drive.google.com/file/d/1FomO05-wiLVFm4W5zNrpZW04_Gj0jwiM/view?usp=sharing
> Qualitatively, the StocCon visualization looks worse than StocCon with prior updating. Both RecRepVCCAP+H and RecRepVCCAP+P tend to form better clusters, and RecRepVCCAP+H+P is clearly better than either +H or +P alone.

---

> ### Comment · Area_Chair1 · 2018-12-10
> **Outstanding review**
>
> As area chair I just wanted to comment that this is an outstandingly thorough, clear, and constructive review.  Thank you.

---

> > ### Author Response · Authors · 2018-12-11
> > **Agreed**
> >
> > Thanks for the detailed and constructive review!

---

### Meta-Review · Area_Chair1 · 2018-12-10
**Many modifications to VAEs with little justification**

**Confidence:** 4
**Recommendation:** Reject

**Metareview:**

This paper heavily modifies standard time-series-VAE models to improve their representation learning abilities.  However, the resulting model seems like an ad-hoc combination of tricks that lose most of the nice properties of VAEs.  The resulting method does not appear to be useful enough to justify itself, and it's not clear that the same ends couldn't be pursued using simpler, more general, and computationally cheaper approaches.